# Renoprotective Effect of KLF2 on Glomerular Endothelial Dysfunction in Hypertensive Nephropathy

**DOI:** 10.3390/cells11050762

**Published:** 2022-02-22

**Authors:** Eunjin Bae, Mi-Yeon Yu, Jong-Joo Moon, Ji-Eun Kim, Saram Lee, Sang-Woong Han, Dong-Jun Park, Yon-Su Kim, Seung-Hee Yang

**Affiliations:** 1Department of Internal Medicine, Institute of Health Science, College of Medicine, Gyeongsang National University, Gyeongsang National University Changwon Hospital, Jinju 52727, Korea; delight7607@naver.com (E.B.); drpdj@naver.com (D.-J.P.); 2Department of Internal Medicine, College of Medicine, Hanyang University Guri Hospital, Guri 11923, Korea; pure8203@gmail.com (M.-Y.Y.); cardion@hanyang.ac.kr (S.-W.H.); 3Department of Internal Medicine, Seoul National University Hospital, Seoul 03080, Korea; jongjoomoon114@gmail.com (J.-J.M.); yonsukim@snu.ac.kr (Y.-S.K.); 4Department of Internal Medicine, Korea University Guro Hospital, Seoul 08308, Korea; beeswaxag@naver.com; 5Transdisciplinary Department of Medicine and Advanced Technology, Biomedical Research Institute, Seoul National University Hospital, Seoul 03080, Korea; hommelee@snu.ac.kr; 6Kidney Research Institute, Seoul National University, Seoul 03080, Korea

**Keywords:** angiotensin type-1 receptor, glomerular endothelial cell, hypertension, Kruppel-like factor 2

## Abstract

Kruppel-like factor 2 (KLF2) regulates endothelial cell metabolism; endothelial dysfunction is associated with hypertension and is a predictor of atherosclerosis development and cardiovascular events. Here, we investigated the role of KLF2 in hypertensive nephropathy by regulating KLF2 expression in human primary glomerular endothelial cells (hPGECs) and evaluating this expression in the kidney tissues of a 5/6 nephrectomy mouse model as well as patients with hypertension. Hypertension-mimicking devices and KLF2 siRNA were used to downregulate KLF2 expression, while the expression of KLF2 was upregulated by administering simvastatin. After 4 mmHg of pressure was applied on hPGECs for 48 h, KLF2 mRNA expression decreased, while alpha-smooth muscle actin (αSMA) mRNA expression increased. Apoptosis and fibrosis rates were increased under pressure, and these phenomena were aggravated following KLF2 knockdown, but were alleviated after simvastatin treatment; additionally, these changes were observed in angiotensin II, angiotensin type-1 receptor (AT1R) mRNA, and interleukin-18 (IL-18), but not in angiotensin type-2 receptor mRNA. Reduced expression of KLF2 in glomerular endothelial cells due to hypertension was found in both 5/6 nephrectomy mice and patients with hypertensive nephropathy. Thus, our study demonstrates that the pressure-induced apoptosis and fibrosis of glomerular endothelial cells result from angiotensin II, AT1R activation, and KLF2 inhibition, and are associated with IL-18.

## 1. Introduction

Kruppel-like factors (KLFs) are members of the zinc finger family of DNA-binding transcription factors [1]. A total of 17 members of the KLF family have been reported, and different functions have been identified depending on the members in the kidney. Endothelial KLF4 has a renoprotective effect against acute kidney injury and KLF4 in podocyte-suppressed proteinuria [2,3]. KLF6 expression in podocytes maintains mitochondrial function and prevents podocyte apoptosis [4], while KLF5 expression prevents podocyte apoptosis via the blockading of ERK/p38 MAPK pathways [5]. KLF15 is a critical regulator of podocyte differentiation and is protective against podocyte injury [6,7,8]. KLF2 mediates the functions of various types of cells, among which its role is particularly crucial in endothelial cells, and it plays a pivotal role in quiescence, shear stress regulation, homeostasis, and vasculogenesis [9,10]. KLF2 is activated by laminar shear stress and inhibits endothelial inflammation [11,12,13] as well as thrombosis [14,15,16]. In contrast, KLF2 expression is reduced by disturbed stress in branched points of blood vessels with atheroprone flow patterns [17,18], resulting in endothelial dysfunction [19,20].

Endothelial dysfunction is a mechanism underlying the development of hypertension [21] and is a predictor of atherosclerosis development and cardiovascular events. It is commonly observed in chronic kidney disease (CKD) and can lead to microalbuminuria, which is independently associated with a reduction in glomerular filtration rate (GFR) and end-stage renal disease [22]. The pathophysiology of hypertensive nephropathy is complex and has not been fully elucidated. Glomerular endothelial dysfunction caused by hypertension can lead to vascular rarefaction, resulting in a reduction in tissue perfusion and consequent ischemia [23,24]. Renin–angiotensin system (RAS) activation plays a key role in end-organ damage via hypertension, which is mediated by angiotensin type-1 receptor (AT1R) expression [25,26,27,28]. AT1R is an angiotensin II receptor, and its activation causes vasoconstriction, the release of aldosterone, and a decrease in urinary sodium excretion in the kidney, thereby raising blood pressure [29,30,31]. We investigated the role of AT1R and KLF2 in glomerular endothelial damage caused by hypertension. Furthermore, we investigated tumor necrosis factor-α (TNFα) and interleukin-18 (IL-18) as proinflammatory cytokines upregulated by angiotensin II. IL-18 is a proinflammatory cytokine elevated in the serum or urine of patients with CKD [32,33,34] and hypertension [35,36,37]. A recent study found a direct association between IL-18 and kidney damage during deoxycorticosterone/salt-induced hypertension in mice [38]. In addition, TNFα causes kidney damage in angiotensin-II-dependent hypertension [39,40] and enhances IL-18 gene expression [41,42].

Previous studies have shown that decreased endothelial KLF2 expression is related to the development of diabetic nephropathy [43] and Alzheimer’s disease [44], with respect to vascular inflammation. Zhong et al. used endothelial-cell-specific KLF2 knockout mice and showed that reduced endothelial KLF2 function could aggravate endothelial injury in diabetic nephropathy [43]. In addition, they found decreased KLF2 expression in mice with unilateral nephrectomy and patients with advanced CKD [45]. Another study showed that the glomerular expression of KLF2 mRNA was lower in patients with thrombotic microangiopathy who have undergone kidney transplantation [46]. However, to date, there have been no studies on the role of KLF2 in hypertensive nephropathy. CKD and hypertension are closely related, both of which can cause and exacerbate each other [47,48,49]. In addition, it is often difficult to know which comes first when hypertension and renal failure are present at the first diagnosis in actual clinical practice. Therefore, we selected a hypertensive CKD animal model, as it reflects real-world clinical practice and elucidates the role of hypertension in CKD. 

Most previous studies have been conducted using animal models, such as those with angiotensin-dependent hypertension and mineralocorticoid–salt hypertension, genetic rats (spontaneously hypertensive rats), nitric-oxide-dependent models, and renovascular models to simulate hypertensive nephropathy [50]. However, experiments that have mimicked increased blood pressure (BP) in cell lines are rare. In vitro studies on hypertension have been conducted using different methods, including increasing atmospheric pressure in an incubator [51] and using a “glomerulus-on-a-chip” microfluidic device [52]. In a previous study, we developed a novel device that applied rotational force to cultured cells and created a hypertensive condition using this method [7]. Using this machine, pressure was applied on glomerular endothelial cells by simulating a hypertensive environment similar to increased BP of the human body, and the cell damage caused by the exerted pressure was evaluated. 

In the present study, we used a novel machine to identify the role of KLF2 in human primary glomerular endothelial cells (hPGECs) in an in vitro model of hypertensive nephropathy. In addition, we evaluated the association between KLF2 and glomerular endothelial cell damage caused by hypertension in 5/6 nephrectomy mice and hypertensive nephropathy patients.

## 2. Materials and Methods

### 2.1. hPGEC Identification

We collected hPGEC-unaffected kidney specimens from nephrectomized patients diagnosed with renal cell carcinoma at the Seoul National University Hospital (Seoul, Korea), as previously described [53]. To identify hPGECs, the cells were stained with anti-CD31 PE (BD Pharmingen, San Jose, CA, USA, 566125). The stained cells were sorted and analyzed using a fluorescence-activated cell-sorting (FACS) Calibur instrument (BD Biosciences, San Jose, CA, USA).

### 2.2. Animal Model of Hypertensive CKD: 5/6 Nephrectomy Mouse Model

Eight-week-old B6 male mice were purchased from the Jackson Laboratory (Bar Harbor, ME, USA) and used in all of the experiments. Surgery was performed under anesthesia with a mixture of xylazine (Rompun; 10 mg/kg; Bayer, Leverkusen, Germany) and Zoletil™ (30 mg/kg; Virbac, Carros, France). A 1:10 dilution of the stock solution in saline was administered intraperitoneally at 0.02 mL/g of body weight. Bilateral dorsal longitudinal incisions were created to expose both kidneys. The lower branch of the right renal artery was ligated with a 10.0 silk suture to produce visible ischemia in approximately one-third of the kidney, and the upper pole of the right kidney was amputated via electrocoagulation [54]. Next, the upper branch of the left kidney artery was ligated, and the lower pole was placed back into the renal fossa via cauterization. Left nephrectomy was performed after 1 week (CKD day 0). The left kidney was removed and the vascular pedicle was ligated to the hilum with a 4–0 silk suture. The time of the left nephrectomy marked the onset of moderate-to-severe renal failure. To induce the damage caused by high BP in the 5/6 nephrectomy mouse CKD model, angiotensin II (0.75 μg/kg/min, Sigma-Aldrich, St. Louis, MO, USA) was mixed with normal saline and infused using an Alzet^®^ implantable subcutaneous osmotic minipump (DURECT Co., Cupertino, CA, USA). The osmotic minipump was inserted into the 5/6 nephrectomy mouse CKD model under isoflurane anesthesia at CKD day 0. The animals were placed in metabolic cages to analyze their systolic BP, body weight, and renal function 20 weeks after the sham surgery or 5/6 nephrectomy. Systolic BP was measured by the tail-cuff method using the CODA system (Kent Scientific Co., Torrington, CT, USA). Serum creatinine was also measured in the blood samples of these mice using Hitachi 747 (Hitachi, Japan). Urine samples were collected for 24 h to determine the protein and creatinine content. The mice were sacrificed at the end of the experiment; the kidneys were harvested. 

### 2.3. Human Samples

Human biopsy specimens of hypertensive nephropathy as well as normal kidneys were collected from the Seoul National University Hospital between April 2010 and April 2019 under a protocol approved by the Institutional Review Board. Hypertensive nephropathy was diagnosed based on typical pathological findings based on renal biopsy and uncontrolled BP [55]. Normal kidneys were diagnosed based on nonspecific findings in the glomeruli or tubules. Clinical data of the patients, including age, sex, body mass index, systolic BP, diastolic BP, estimated GFR (eGFR), and urine protein/creatinine ratio, were obtained from a review of the medical records. These data were assessed at the time of the kidney biopsy. To compare the expression of KLF2 in glomerular endothelial cells according to BP, we divided the patients into normal controls and patients with hypertensive nephropathy. High BP was defined as greater than or equal to 140/90 mmHg according to the Joint National Committee 7 (JNC7) guideline [56].

### 2.4. Hypertensive Injury in hPGECs Induced via a Pressurizing Device

To cause hypertensive injury, hPGECs were cultured in 6-well culture plates (2 × 10^5^ cells/well); four culture plates were mounted on the proposed pressurizing system for the experiments. The pressurizing system was placed inside a 5% CO_2_ incubator at 37 °C during the experiments. Optimal pressure conditions were evaluated for the induction of fibrosis in hPGECs. Three operating pressures were assessed, namely static, 4 mmHg, and 8 mmHg, similar to a previous study that assessed podocyte fibrosis [7]. We determined the pressure to induce fibrosis in hPGECs to be 4 mm Hg in this experiment. 

### 2.5. Histology and Immunohistochemistry

Masson’s trichrome staining (Sigma-Aldrich), and Sirius red staining (Abcam, Cambridge, UK) were performed to evaluate glomerular sclerosis. For the immunohistochemical assays, paraffin-embedded kidneys were cut into 4-μm-thick slices, deparaffinized, and hydrated using xylene and ethanol. Endogenous streptavidin activity was blocked using 3% hydrogen peroxide. The deparaffinized sections were stained with an anti-KLF2 antibody for kidneys and then incubated with horseradish-peroxidase-conjugated anti-mouse IgG (DAKO, Carpinteria, CA, USA, K3954). Next, 3,3′-diaminobenzidine tetrahydrochloride (Sigma-Aldrich) was used for immunohistochemical detection. Finally, all samples were counterstained with Mayer’s hematoxylin (Sigma-Aldrich) and evaluated under a light microscope (DFC-295; Leica, Mannheim, Germany). For each sample, five fields (X400) were randomly selected, and blue- (Masson’s trichrome staining) as well as brown-stained areas (immunohistochemistry) that reflected kidney fibrosis were quantified using computer-based morphometric analysis (Qwin 3; Leica). Scoring was performed in a blinded manner using the mean values of the positive areas (%).

### 2.6. Quantitative Reverse Transcription Polymerase Chain Reaction (qRT-PCR) Analysis

Total RNA was extracted from the cells, and mRNA levels of the target genes were assessed using qRT-PCR. Briefly, total RNA was extracted from hPGECs using an RNeasy kit (Qiagen GmbH, Hilden, Germany), and 50 ng of total RNA were reverse-transcribed using oligo-d (T) primers and AMV-RT Taq polymerase (Promega, Madison, WI, USA). qRT-PCR was performed using the SYBR Green method and primers for alpha-smooth muscle actin (αSMA), KLF4, KLF2, transforming growth factor (TGFβ), AT1R, angiotensin type-2 receptor (AT2R), and glyceraldehyde-3-phosphate dehydrogenase (GAPDH) (Applied Biosystems, Foster City, CA, USA). The PCR data were analyzed using the Applied Biosystems PRISM 7500 sequence detection system. Relative quantification was performed using the 2^−ΔΔC_T_^ method [57]. GAPDH was used as a loading control, and mRNA expression levels were normalized to those of GAPDH. All the experiments were performed in triplicate. The PCR primers used for qRT-PCR are listed in Table 1.

### 2.7. Confocal Microscopy Examination

Deparaffinized sections were stained with immunofluorescence antibodies against KLF2 (Novus, MAB5466-SP) as well as 4′,6-diamidino-2-phenylindole (DAPI; Sigma-Aldrich, D9542) and incubated with a blocking reagent overnight at 4 °C. After selectively applying Alexa-Fluor-488-conjugated goat anti-mouse antibodies (Invitrogen, Carlsbad, CA, USA, A-11001) as secondary antibodies, the deparaffinized sections were counterstained with DAPI for an additional 5 min. KLF2 expression was visualized using immunofluorescence staining with a confocal microscope (Leica TCS SP8 STED CW instrument, Leica) and MetaMorph software (version 7.8.10, Universal Imaging, Downingtown, PA, USA). The sections were evaluated in a blind and random manner, and images were quantified by counting the positively stained cells in five separate high-power fields (X600) per well using Image-Pro Plus software (Media Cybernetics, Bethesda, MD, USA). 

### 2.8. Regulation of KLF2

#### 2.8.1. Upregulation of KLF2 Expression via Simvastatin Treatment

Simvastatin (Simvastatin^®^, Sigma-Aldrich, 79902-63-9) was used for upregulating KLF2 expression in pressure-induced kidney injury in this study [58,59]. hPGECs were simultaneously cultured for 48 h with or without different doses of simvastatin (1 μM and 10 μM; dissolved in 2% dimethyl sulfoxide (Sigma-Aldrich) in saline). 

#### 2.8.2. Downregulation of KLF2 Expression Using Pressurizing Devices that Employ Rotational Force

We used pressurizing devices to apply pressure (4 and 8 mmHg) to the cultured hPGECs for 48 h. The change in KLF2 expression with varying pressure was confirmed and analyzed. The pressurizing device used was described in a previous study [7]. 

#### 2.8.3. Transfection of siRNA in hPGECs

The specific small interfering RNA (siRNA) targeting KLF2 was purchased from Santa Cruz Biotechnology (Dallas, TX, USA, sc-35818). To inhibit KLF2 activity, siRNA was transfected into hPGECs, as described in the manufacturer’s protocol. We seeded 2 × 105 hPGECs per well in a microtiter plate (NUNC A/S, Denmark) in 2 mL of antibiotic-free normal growth medium supplemented with fetal bovine serum for 24 h prior to transfection. The cells were grown to 60–80% confluence at 37 °C in a CO_2_ incubator. The siRNA duplex solution was diluted with a transfection reagent. After one day of transfection, the cells were then pressurized with the designed pressurizing device for 48 h at 37 °C. 

### 2.9. FACS

hPGECs were washed with cold phosphate-buffered saline, resuspended in 100 μL of binding buffer, stained with 5 μL of FITC-conjugated annexin V (10 mg/mL) and 10 μL of propidium iodide (PI) (50 mg/mL), and incubated for 30 min at room temperature in the dark. The proportion of apoptotic and necrotic cells was measured using an annexin V/PI FITC apoptosis kit (BD Biosciences) via flow cytometry after pressurization (4 mmHg for 48 h) with or without simvastatin (1 μM, 10 μM) and transfection with KLF2 siRNA (2 μM). Annexin V-FITC binds to phosphatidylserine, which is translocated from the inner to the outer plasma membrane during early apoptosis. PI is a cell-impermeable nuclear dye that is excluded by viable and early apoptotic cells. However, it is taken up by necrotic or late apoptotic cells, resulting in red fluorescence. In an annexin assay, cells that are annexin-negative and DNA-dye-positive are often considered necrotic. FITC-labeled fibronectin was used instead of fibronectin. FACS analysis was performed to measure the antifibrotic effect of simvastatin, and the acquired data were analyzed with BD FACSDiva (version 8.0; BD Biosciences). 

### 2.10. ELISA

The serum TNFα (DuoSet; R&D Systems, Minneapolis, MN, USA, DY210), IL-18 (R&D Systems, DY8936-05), and angiotensin II (CUSABIO, China, Wuhan, CSB-E04493h) levels were measured by an enzyme-linked immunosorbent assay (ELISA) according to the manufacturer’s instructions. The concentration was estimated from a standard curve and expressed as the mean ± standard error of the mean (SEM).

### 2.11. Western Blot Analysis

Proteins were separated by 10% sodium dodecyl sulfate–polyacrylamide gel and transferred onto Immobilon-FL 0.4 μm polyvinylidene difluoride membranes (Millipore, Darmstadt, Germany). Primary antibodies targeting p53 (Santa Cruz, sc126), fibronectin (Abcam, ab2413), α-SMA (Abcam, ab5694), AT1R (Santa Cruz, sc515884), KLF2 (Abcam, ab203591), β-actin (Sigma-Aldrich, A1978), and TGFβ (Abcam, ab66043) were used. After the membranes were treated with primary antibodies, anti-rabbit (1:5000; Cell Signaling Technology, Danvers, MA, USA) and anti-mouse (1:5000 for β-actin; Cell Signaling Technology) secondary antibodies were used as appropriate. Using an Image Quant LAS 4000 mini (GE Healthcare), the intensities of the immunoblot band were visualized and captured, and ImageJ (National Institutes of Health, Bethesda, MD, USA) was used to conduct a densitometric analysis of protein expression. The expression levels of the target molecules were normalized to β-actin expression.

### 2.12. Statistical Analysis

We used a one-way analysis of variance for continuous variables and a Chi-square test for proportions. Data are expressed as mean ± standard deviation or median with range, or frequency wherever indicated. A Student’s *t*-test was performed to compare fold changes in qRT-PCR experiments. Statistical analyses were performed using SPSS version 22 (IBM software, Armonk, NY, USA) and GraphPad Prism 8.0 (GraphPad Software, Inc., La Jolla, CA, USA). Statistical significance was set at *p* < 0.05.

## 3. Results

### 3.1. Effects of Mechanical Pressurization by Rotational Force on Glomerular Endothelial Cells and Reduction in KLF2 Expression

We previously developed a pressurizing device that mimics high BP in human vessels responsible for causing hypertensive injury, in vitro. In addition, we determined the optimum pressure to maintain cell viability while injuring hPGECs using this device. The viability of hPGECs cultured at a pressure of 4 mmHg was more than 80%, which was not significantly different from that of cells cultured in static conditions, but the viability decreased at a pressure exceeding 4 mmHg [7]. 

To determine whether glomerular endothelial cell damage was caused by pressure and to verify the change in the levels of KLF2 caused by pressure, qRT-PCR was performed. After the application of 4 mmHg pressure to hPGECs for 48 h, the mRNA expression of αSMA, a fibrosis marker, was increased compared to that under static conditions (Figure 1A). Interestingly, compared to the static conditions, the mRNA expression of KLF4, a representative transcription factor which also has protective effects on endothelial cells, did not change under pressure, but KLF2 mRNA expression decreased significantly (*p* < 0.01) (Figure 1B,C). These results suggest that the pressure-induced fibrosis of glomerular endothelial cells is mediated by KLF2. Based on these findings, this study focused on the role of KLF2 in endothelial cell injury caused by hypertension.

### 3.2. Pressure-Induced Apoptosis and Its Change through KLF2 Upregulation or Knockdown

The annexin V-FITC/PI staining assay and flow cytometry were used to determine the distribution of cells belonging to four different apoptotic stages: viable cells, early apoptosis, late apoptosis, and necrotic cells. The number of apoptotic cells increased with increasing pressure and was significantly reduced by simvastatin treatment. In early apoptosis, these changes were observed with low- and high-dose simvastatin (1 μM, 10 μM). Late apoptosis was not recovered at low simvastatin doses, but decreased at high simvastatin doses. A pressure of 4 mmHg increased the degree of cell necrosis, but the number of pressure-induced necrotic cells was significantly reduced in the case of both low- and high-dose simvastatin treatment (Figure 2A). As expected, the mRNA expression of KLF2 was decreased by pressurization, and this change was alleviated via simvastatin treatment (Figure 2B). To evaluate whether pressure-induced apoptosis was associated with KLF2, we performed KLF2 upregulation via simvastatin treatment as well as its knockdown by KLF2 siRNA transfection in hPGECs under pressurized conditions (Figure 2C). Compared to the group subjected only to pressure, the proportion of apoptotic cells was significantly lower in the simvastatin treatment group (17.8 ± 0.9% under 4 mmHg pressure, 14.7 ± 1.0% in 4 mmHg pressure with 10 µM simvastatin treatment, *n* = 5/group, *p* < 0.001) and higher in the siRNA-transfected group (21.6 ± 1.5% at 4 mmHg pressure with 2 μM KLF2 siRNA, *n* = 5/group, *p* < 0.001). These results suggest that the apoptosis of glomerular endothelial cells is mediated by KLF2.

### 3.3. Induction of Fibrotic Markers by Pressure

We examined the role of KLF2 in pressure-induced fibrosis in hPGECs. After the application of static, 4, and 8 mmHg pressure, the expression of fibronectin was determined via flow cytometry analysis. Compared to in the static conditions, fibronectin was increased under pressure, with a higher increase at 8 mmHg compared to 4 mmHg. The proportions of fibronectin-positive cells were 25.2 ± 3.6% under static conditions, 59.6 ± 5.6% under 4 mmHg pressure, and 69.2 ± 3.0% under 8 mmHg pressure (Figure 3A). To determine the effects of KLF2 on glomerular endothelial cells we conducted KLF2 upregulation via simvastatin treatment in addition to KLF2 knockdown via KLF2 siRNA transfection. Pressure-induced fibrosis was significantly decreased via KLF2 upregulation and increased via KLF2 knockdown, as compared to the application of pressure only: the proportion of fibronectin-positive cells was 59.6 ± 5.6% under 4 mmHg pressure, 45.7 ± 5.7% under 4 mmHg pressure with 10 μM simvastatin treatment, and 71.4 ± 22.9% in the 4 mmHg pressure with 2 μM KLF2 siRNA (*n* = 4/group, *p* < 0.01) (Figure 3B). Additionally, we evaluated the changes in the mRNA expression of TGF-β and fibronectin. Compared to the pressure-only conditions, the expression of TFG-β mRNA was significantly lower in the 10 µM simvastatin treatment group and higher in the siRNA-transfected group (Figure 3C). Trends in the expression of fibronectin mRNA were similar to that of TGF-β mRNA (Figure 3D). These results suggest that the pressure-induced fibrosis of glomerular endothelial cells is mediated by KLF2. 

### 3.4. Changes in the Levels of Angiotensin II, AT1R, AT2R, and Proinflammatory Markers in Response to KLF2 Upregulation and Knockdown under Pressurized Conditions

We investigated whether angiotensin II, one of the key factors in the mechanism of hypertensive nephropathy, is produced in hPGECs through mechanical pressurization via rotational force. The level of angiotensin II that had been enhanced under 4 mmHg pressure was significantly reduced by 10 μM simvastatin treatment, but was boosted further by 2 μM KLF2 siRNA transfection (Figure 4A). Additionally, we evaluated the changes in the mRNA expression of AT1R and AT2R through qRT-PCR to investigate the AT1R and AT2R pathways involved in pressure-induced glomerular endothelial cell injury associated with KLF2. AT1R mRNA expression, which was increased under 4 mmHg pressure, was significantly decreased by 10 µM simvastatin treatment, but was further increased by 2 μM KL2 siRNA transfection (Figure 4B). Under 10 μM simvastatin treatment, AT2R mRNA expression showed a trend similar to that of AT1R mRNA. However, no significant change was observed in this case by siRNA transfection, contrasting with that of AT1R mRNA expression (Figure 4B). These results suggest that KLF2-associated pressure-induced glomerular cell injury is regulated more dominantly by AT1R than by AT2R. Furthermore, we evaluated the changes in the proinflammatory markers, IL-18, and TNFα via an ELISA under the same conditions. The production of IL-18 was upregulated by pressurization and significantly aggravated by KLF2 siRNA pretreatment, but simvastatin treatment ameliorated these changes (Figure 4C). Changes in TNFα due to pressurization with KLF2 siRNA pretreatment were similar to those of IL-18, but no significant change was observed with simvastatin treatment (Figure 4C). These results suggest that IL-18 contributes to KLF2-mediated glomerular endothelial cell damage caused by hypertension. The effect of KLF2 siRNA was evaluated by Western blot, and the transfection of KLF2 siRNA under static conditions reduced KLF2 expression by 50% (Figure 4D). In addition, we evaluated the changes in apoptosis (p53), fibrosis (fibronectin, αSMA, and TGFβ) markers, and AT1R under pressurized conditions at the protein level using Western blot. Compared to the pressure-only conditions, the expression of p53 proteins was significantly lower in the 10 µM simvastatin treatment and higher in the siRNA-transfected group (Figure 4E). Trends in the expression of fibronectin, αSMA, TGFβ, and AT1R were similar to that of the p53 protein (Figure 4E).

### 3.5. Hypertensive Kidney Injury Reduces KLF2 Expression in 5/6 Nephrectomy Mouse Model

Eight-week-old B6 male mice were used to examine the effect of chronic hypertension on the expression of KLF2 in glomerular endothelial cells. Animals in the hypertensive nephropathy group underwent 5/6 nephrectomy and angiotensin II infusion via the osmotic minipump to induce chronic hypertensive injury and fibrosis. Histological examination showed glomerular fibrosis, coarse vacuolization, and the loss of the brush border of kidney tubular epithelial cells. Glomerula KLF2 expression was significantly decreased in 5/6 nephrectomized mice (13.3 ± 8.5% in the sham group and 4.8 ± 2.3% in the 5/6 nephrectomy group, n = 6 for each group, *p* < 0.01) (Figure 5A). Twenty weeks after the 5/6 nephrectomy, an increase in systolic BP from 116.0 ± 9.06 mmHg to 172.3 ± 10.5 mmHg, serum creatinine from 0.3 ± 0.1 mg/dl to 1.1 ± 0.2 mg/dl, urine protein/creatinine ratio from 0.9 ± 0.3 mg/mg to 9.3 ± 2.5 mg/mg, and a decrease in weight from 24.2 ± 0.8 g to 20.1 ± 1.4 g were observed, as compared with the sham-operated mice (Figure 5B). Thus, hypertension appears to contribute to kidney injury and reduce KLF2 expression in vivo. 

### 3.6. Decrease in KLF2 Expression in Glomerular Endothelial Cells in Patients with Hypertensive Nephropathy

We performed an immunohistochemical assay of KLF2 in biopsied kidney tissue from both the normal control and hypertensive nephropathy groups. The baseline characteristics of the patients are shown in Table 2. Patients with hypertensive nephropathy showed significantly higher systolic BP, higher diastolic BP, lower eGFR, and higher proteinuria than normal controls. There were no statistically significant differences in age, sex, or body mass index (BMI). The hypertensive nephropathy group had prescribed a higher proportion of antihypertensive drugs compared to the normal group, and among the antihypertensive drugs, calcium channel blockers were the most common, and the second were angiotensin-converting enzyme inhibitors (ACEIs) or angiotensin II receptor blockers (ARBs). KLF2 expression in biopsied kidney tissues is shown in Figure 6. Immunohistochemical staining showed that KLF2 expression was decreased in kidneys with hypertensive nephropathy compared to that in normal kidneys (*p* < 0.05) (Figure 6A). The percentage of KLF2 expression was significantly lower in hypertensive nephropathy (1.5 ± 0.9%, *n* = 9) than in the control group (3.5 ± 2.5%, *n* = 8, *p* < 0.05).

## 4. Discussion

In this study, we found that angiotensin II expression was increased in hPGECs through mechanical pressurization via rotational force, which led to increases in apoptosis, fibrosis, and proinflammatory markers via the AT1R-dominant pathway. These results were alleviated by KLF2 upregulation and worsened by KLF2 knockdown, suggesting that pressure-induced apoptotic and fibrotic injuries in hPGECs are mediated by KLF2. In the mouse model, we also found that the expression of KLF2 in glomerular endothelial cells was decreased by hypertensive kidney injury. Furthermore, the expression of KLF2 in glomerular endothelial cells was lower in patients with high BP than in patients with normal BP. This also suggests that the expression of KLF2 in glomerular endothelial cells is specifically affected by BP. 

We recently developed a novel machine that induces hypertensive injury in podocytes in vitro [7]. Podocytes were damaged and fibrous owing to the pressure generated by the machine. In this study, since endothelial cell damage is a key mechanism underlying hypertensive nephropathy, we used glomerular endothelial cells as target cells of hypertensive nephropathy. In addition, for the first time, using this machine to apply pressure also increases the expression of angiotensin II in hPGECs, which is linked to fibrosis and apoptosis. The activated angiotensin–aldosterone system in hypertension causes endothelial damage directly [60] or indirectly by inhibiting endothelial cell regeneration and promoting inflammatory pathways [61]. Endothelial injury itself causes hypoxic damage [62], which is an important mechanism contributing to hypertensive nephropathy and fibrosis [63]. Damage to glomerular endothelial cells, one of the components of the glomerular filtration barrier, results in albuminuria, podocyte injury, and CKD progression [64,65]. Our results also showed that the apoptosis and fibrosis of glomerular endothelial cells were increased by hypertensive injury, suggesting that a decrease in the KLF2 expression of glomerular endothelial cells may be one of the mechanisms of hypertensive nephropathy. 

We demonstrated that pressure-induced apoptosis and fibrosis are mediated by KLF2 in glomerular endothelial cells. Consistent with our study, previous studies have shown that the upregulation of KLF2 has an endothelial protective effect against apoptosis [66,67]. Another study showed that the decreased expression of KLF2 in endotoxic mice resulted in increased endothelial apoptosis, which was alleviated by KLF2 upregulation [68]. In addition, previous studies have shown endothelial antifibrotic effects through upregulation of KLF2 in cirrhotic rat liver [67] and cardiac fibrosis [69]. However, no studies have evaluated the role of KLF2 in hypertensive nephropathy using hPGECs. We performed in vitro experiments by applying pressure to hPGECs using a unique hypertension-mimicking machine, and we demonstrated the expression of KLF2, which is reduced by hypertension in mice and humans. In addition, increased proteinuria, decreased GFR, and the fibrosis of glomerular endothelial cells due to hypertension have been observed in human and murine models. This shows that a decrease in the expression of KLF2 is related to kidney injury caused by hypertension. Furthermore, we evaluated the association of pressure-induced fibrosis with KLF2 and angiotensin II receptors. Angiotensin II, a key player in hypertension, causes vasoconstriction, inflammation, oxidative stress, apoptosis, and fibrosis through AT1R, but AT2R negatively regulates AT1R-dependent signaling [70,71]. In our study, the pressure-induced damage of glomerular endothelial cells was mediated by KLF2, which mainly occurs through angiotensin II-AT1R, but not AT2R. The result of simvastatin mitigating the increase in AT2R mRNA expression via pressurization is presumed to be due to other renoprotective effects of simvastatin rather than KLF2 upregulation [72]. 

Additionally, glomerular endothelial injury caused by pressurization increased the level of IL-18, a proinflammatory cytokine, in a KLF2-dependent manner. Previous studies have shown that angiotensin II activates IL-18-induced inflammatory genes [73] and participates in atherogenesis [74]. As in previous studies, IL-18 is an important cytokine that induces inflammation, which may lead to target organ damage. These findings suggest that angiotensin II-AT1R-KLF2-IL-18 is a mechanism of the apoptosis and fibrosis of hypertensive nephropathy. 

This study has several limitations. The mechanism underlying the association between a reduction in the level of KLF2, apoptosis, and fibrosis in hypertensive kidney injury has not been identified. Furthermore, simvastatin has anti-inflammatory effects, and therefore it cannot completely explain the KLF2-specific renoprotective effect. The mechanism of hypertensive nephropathy is complex and cannot be explained solely by endothelial cell damage. Thus, further studies are required, not only on glomerular endothelial cells but also on changes in other cells, such as tubular epithelial cells in the kidney. In addition, the hypertension-mimicking machine requires further optimization to function in fluidic environments similar to blood flow within blood vessels. 

Here, we demonstrated that KLF2-angiotensin II-AT1R-IL-18 might be an important mediator of apoptosis and fibrosis in the pathophysiology of endothelial cell damage caused by hypertension. Our study suggests that the upregulation of KLF2 expression is a potential therapeutic strategy against hypertensive nephropathy. We aim to conduct a study in the future to investigate the upregulation of KLF2 expression and the mechanism underlying its protective effect on hypertensive nephropathy using a fluidic environment that is more similar to blood flow. 

## Figures and Tables

**Figure 1 cells-11-00762-f001:**
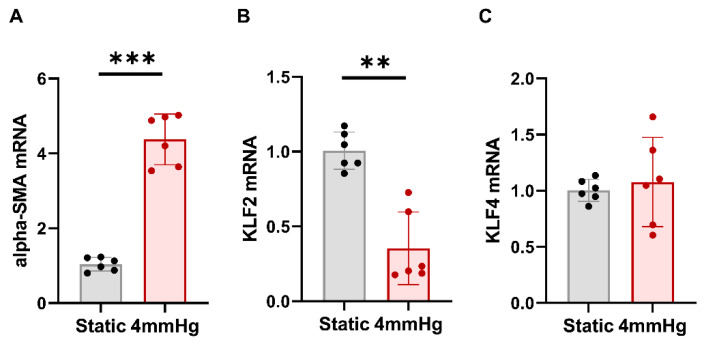
Mechanical pressure on human primary glomerular endothelial cells significantly increases αSMA expression and decreases KLF2 expression, while it has no significant effect on KLF4 expression. qRT-PCR of (**A**) αSMA, (**B**) KLF2, and (**C**) KLF4 was performed under static and pressurized conditions (4 mmHg for 48 h) to confirm the effect of mechanical pressure on human primary glomerular endothelial cells (*n* = 6 per group; ** *p* < 0.01, *** *p* < 0.001).

**Figure 2 cells-11-00762-f002:**
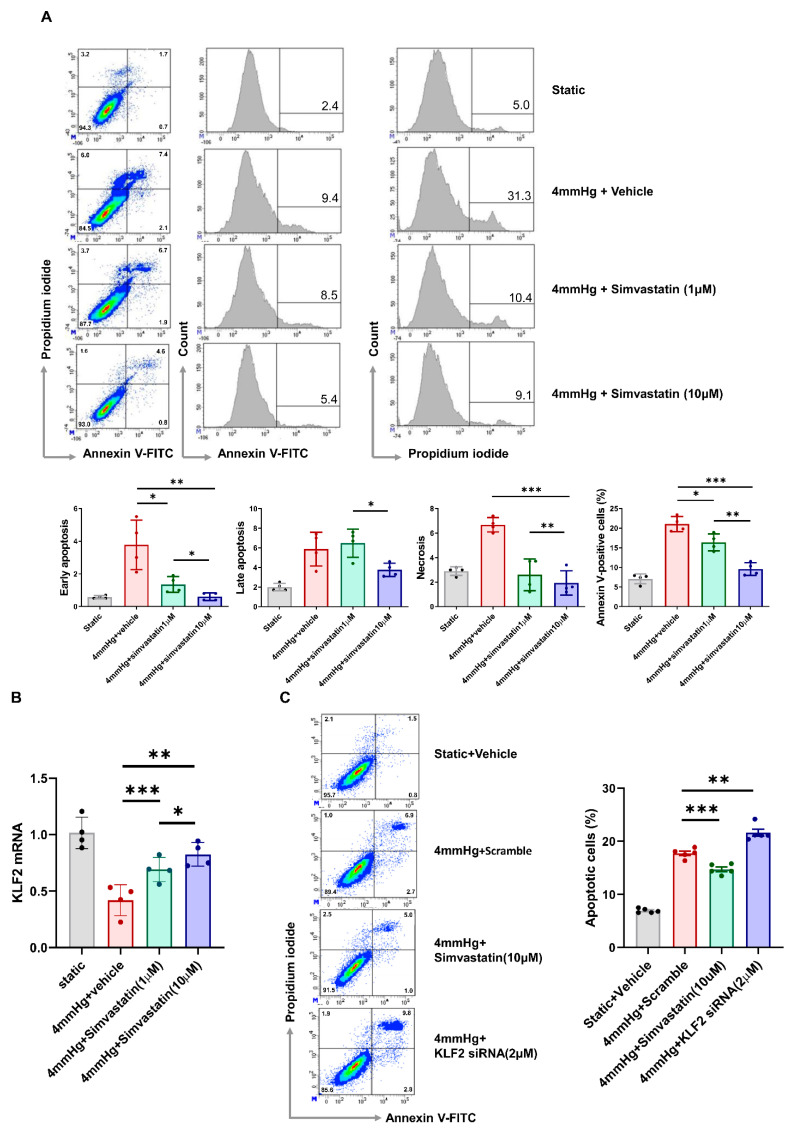
Simvastatin treatment alleviates pressure-induced apoptosis and KLF2 inhibition aggravates pressure-induced apoptosis in primary glomerular endothelial cells. (**A**) Flow cytometry images of apoptosis and necrosis in primary glomerular endothelial cells under static and pressurized conditions (4 mmHg for 48 h) with/without 1 μM and 10 μM simvastatin treatment. Proportions (mean ± SEM) of early apoptotic, late apoptotic, necrotic, and annexin-V-FITC/PI-positive cells under static and pressurized conditions (4 mmHg for 48 h) with/without 1 μM and 10 μM simvastatin treatment (*n* = 4 per group; * *p* < 0.05, ** *p* < 0.01, and *** *p* < 0.001). (**B**) Proportion of KLF2-expressing cells under static and pressurized conditions (4 mmHg for 48 h) with/without 1 μM and 10 μM simvastatin treatment (*n* = 4 per group; * *p* < 0.05, ** *p* < 0.01, and *** *p* < 0.001). (**C**) Proportion of apoptotic cells (annexin-V-FITC/PI-positive cells) under static and pressurized conditions (4 mmHg for 48 h) with 10 μM simvastatin and 2 μM KLF2 siRNA (*n* = 5 per group; ** *p* < 0.01, *** *p* < 0.001).

**Figure 3 cells-11-00762-f003:**
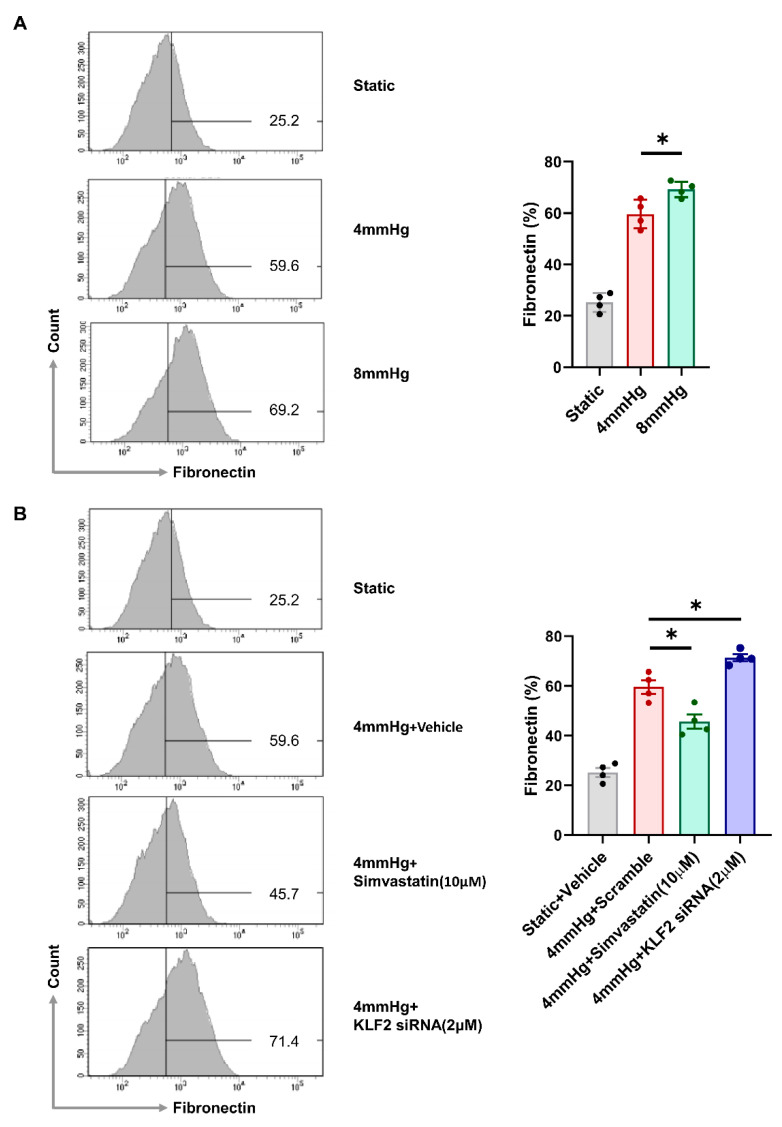
Pressure-induced fibrosis in primary glomerular endothelial cells is alleviated following the upregulation of KLF2 expression and aggravated by KLF2 inhibition. (**A**) Flow cytometry analysis of fibronectin-positive primary glomerular endothelial cells under static and pressurized conditions (4 mmHg, 8 mmHg for 48 h, *n* = 4 per group; * *p* < 0.05). (**B**) Proportion of fibronectin-expressing cells under static and pressurized conditions (4 mmHg for 48 h) with 10 μM simvastatin and 2 μM KLF2 siRNA treatment (*n* = 4 per group; * *p* < 0.05). qRT-PCR of (**C**) TGFβ, (**D**) fibronectin under static and pressurized conditions (4 mmHg for 48 h) as well as the pressurized conditions with 10 μM simvastatin or 2 μM KLF2 siRNA treatment *(n* = 12 per group; * *p* < 0.05, *** *p* < 0.001).

**Figure 4 cells-11-00762-f004:**
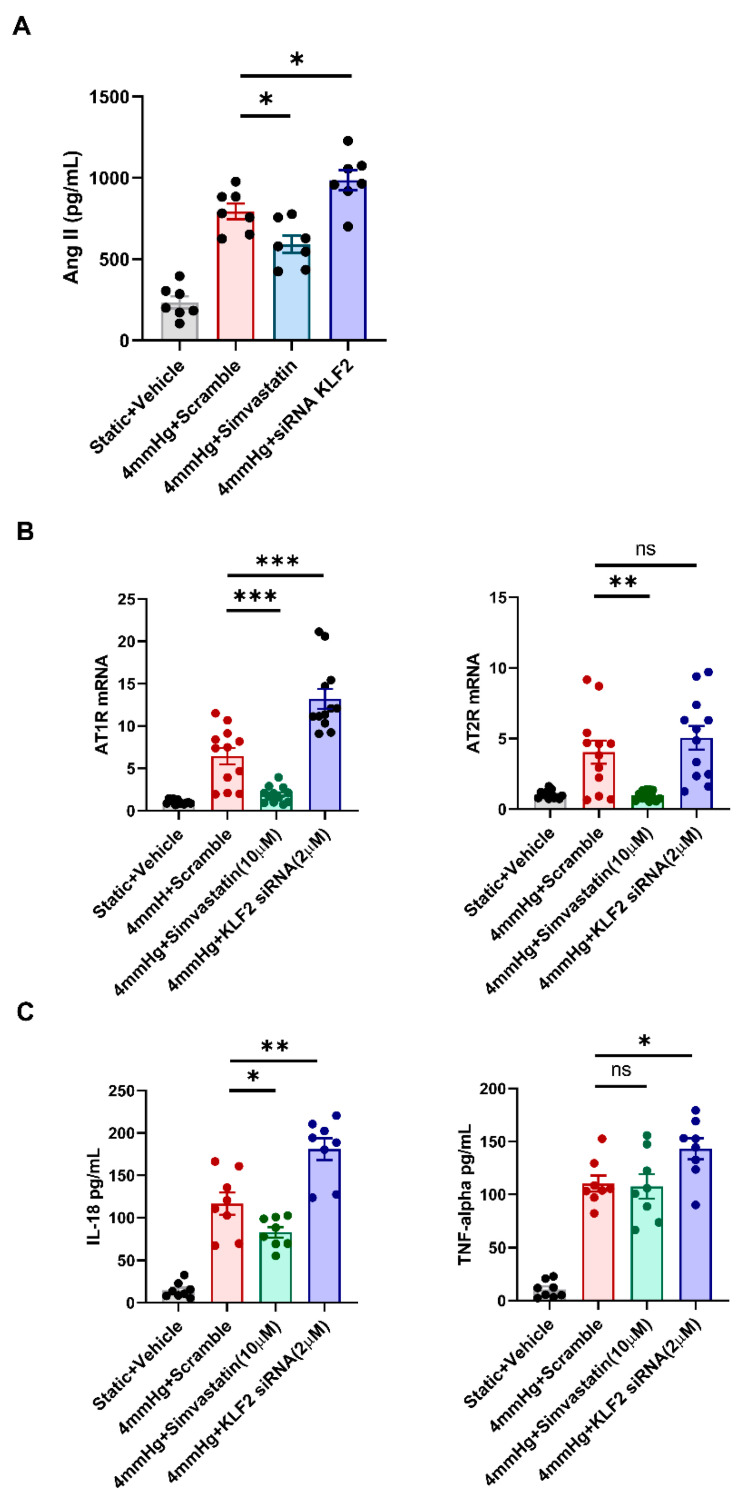
Pressure-induced injury in primary glomerular endothelial cells is regulated dominantly by angiotensin II and AT1R, and promotes inflammation. (**A**) ELISA of angiotensin II under static and pressurized conditions (4 mmHg for 48 h) as well as pressurized conditions with 10 μM simvastatin and 2 μM KLF2 siRNA treatment (*n* = 7 per group; * *p* < 0.05). (**B**) qRT-PCR of angiotensin II type-1 receptor and angiotensin II type-2 receptor under static and pressurized conditions (4 mmHg for 48 h) as well as pressurized conditions with 10 μM simvastatin and 2 μM KLF2 siRNA treatment (*n* = 12 per group; ** *p* < 0.01, *** *p* < 0.001). (**C**) ELISA of IL-18 and TNFα in culture media under static and pressurized conditions (4 mmHg for 48 h) as well as pressurized conditions with 10 μM simvastatin and 2 μM KLF2 siRNA treatment (*n* = 12 per group; * *p* < 0.05, ** *p* < 0.01). (**D**) Western blot of KLF2 under static conditions with 2 μM KLF2 siRNA treatment (*n* = 6 per group; *** *p* < 0.001). (**E**) Western blot of fibronectin, αSMA, AT1R, p53, and TGFβ under static and pressurized conditions (4 mmHg for 48 h) as well as pressurized conditions with 10 μM simvastatin and 2 μM KLF2 siRNA treatment (*n* = 4 per group; * *p* < 0.05, ** *p* < 0.01, and *** *p* < 0.001).

**Figure 5 cells-11-00762-f005:**
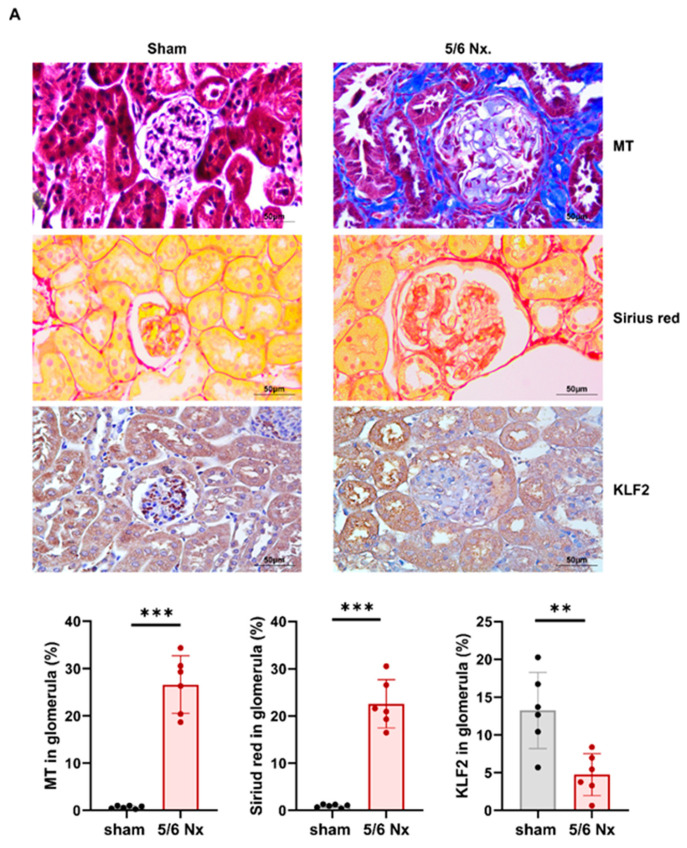
Reduction in KLF2 expression, deterioration of kidney function, occurrence of proteinuria and hypertension, and weight loss in 5/6 nephrectomy mice. (**A**) Representative images of Masson’s trichrome staining, and Sirius red staining for the fibrosis of kidney tissue in mice subjected to sham surgery and mice with 5/6 nephrectomy (*n* = 6, *** *p* < 0.001). Immunohistochemical labeling of KLF2 in mice subjected to sham surgery and mice with 5/6 nephrectomy (*n* = 6, ** *p* < 0.01). (**B**) Plasma creatinine concentrations, urine protein/creatinine ratio, body weight, and systolic BP (mean ± SEM) in the sham as well as 5/6 nephrectomy group 20 weeks after the surgery (*n* = 6, *** *p* < 0.001).

**Figure 6 cells-11-00762-f006:**
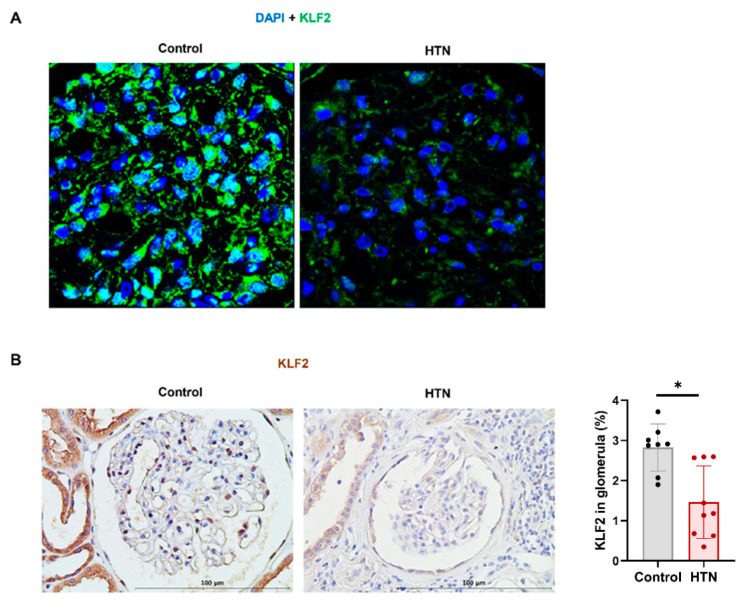
High blood pressure significantly reduces the expression of KLF2 in glomerular endothelial cells. (**A**) Representative immunofluorescence image of KLF2 for patients with normal kidneys and patients with hypertensive nephropathy. (**B**) Immunohistochemical labeling of KLF2 in controls (normal kidneys with normal BP) and hypertension (hypertensive nephropathy). Proportions of KLF2-positive cells (mean ± SEM) in each group (control = 9, hypertension = 8, * *p* < 0.05).

**Table 1 cells-11-00762-t001:** Primers for humans.

Targets	Sense (5→3)	Antisense (5→3)
GAPDH	TCGACAGTCAGCCGCATCT	CCGTTGACTCCGACCTTCA
αSMA	GATGGGCATCTATCAGATAC	AAGCATTTCTGATGGTGATG
KLF2	GCAAGACCTACACCAAGAGTTCG	CATGTGCCGTTTCATGTGC
KLF4	AGGGGGTGACTGGAAGTTGT	TTGCACATCTGAAACCACAG
TGFβ	CCCAGCATCTGCAAAGCTC	GTCAATGTACAGCTGCCGCA
Fibronectin	CCACCCCCATAAGGCATAGG	GTAGGGGTCAAAGCACGAGTCATC
AT1R	CCGCATTTAACTGCTCACACA	ATCATGTAGTAGAGAACAGGAATTGCTT
AT2R	CGGAATTCATGAGCTGCGTTAATCC	AACTGCAGTTAAGACACAAAGGTCTCCA

**Table 2 cells-11-00762-t002:** Baseline characteristics of enrolled patients.

Variables	Total (*n* = 17)	Normal (*n* = 9)	Hypertensive Nephropathy (*n* = 8)	*p*-Value
Age	43.6 ± 12.3	40.4 ± 10.8	47.3 ± 13.4	0.115
Men (%)	12 (63.2)	4 (40.0)	8 (88.9)	0.027
BMI (kg/m^2^)	25.1 ± 3.3	25.0 ± 3.4	25.2 ± 3.4	0.894
Systolic blood pressure (mmHg)	151.0 ± 47.0	116.9 ± 6.2	188.9 ± 43.1	<0.001
Diastolic blood pressure (mmHg)	95.6 ± 31.1	76.4 ± 7.2	116.9 ± 33.9	0.002
eGFR (mL/min/1.73 m^2^)	72.7 ± 45.0	107.4 ± 26.0	29.3 ± 13.6	<0.001
Urine protein–creatinine ratio (g/gCr)	0.5 (0.1, 1.8)	0.3 (0.1, 1.9)	1.1 (0.2, 2.6)	0.247
KLF2 expression in glomerular endothelial cells (%)	2.5 ± 2.1	3.5 ± 2.5	1.5 ± 0.9	0.037
Antihypertensive drug (%)	9 (52.9)	1 (11.1)	8 (88.9)	<0.001
ACEI or ARB (%)	7 (41.2)	1 (11.1)	6 (75.0)	0.008
Beta blocker	4 (23.5)	0 (0.0)	4 (50.0)	0.015
Calcium channel blocker	7 (41.2)	0 (0.0)	7 (87.5)	<0.001
Diuretics	1 (5.9)	0 (0.0)	1 (12.5)	0.274

The data are represented by mean ± SD and proportion (%).

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
