# Peer review of "Renoprotective Effect of KLF2 on Glomerular Endothelial Dysfunction in Hypertensive Nephropathy"

_cells, 2022, doi:10.3390/cells11050762_

Round 1

Reviewer 1 Report

The authors have satisfactorily addressed all my comments. They have included new figures and made changes in the text as requested. I have no further comments or queries.

Author Response

We really appreciate your comment. And we will be honored to be published in this journal.

Reviewer 2 Report

The new version of the manuscript was improved, considering new results and conclusions.

The only point not addressed by authors was the representative images for fibrosis in panel 5a. I think that once that these images can be improved, that manuscript can be finally accepted for publication. 

Author Response

Thank you for the comments. We changed the representative images for fibrosis in figure 5a as recommended.

This manuscript is a resubmission of an earlier submission. The following is a list of the peer review reports and author responses from that submission.

Round 1

Reviewer 1 Report

The authors present to KLF2 as a protein that modulates apoptosis and regulates some pro-inflammatory and pro-fibrotic markers in human primary glomerular endothelial cells (hPGECs) submitted to an In Vitro system of pressure by rotational force. This was complemented by an experimental model of chronic kidney disease in mice, and by the KFL2 measurement in renal biopsies of patients with hypertensive nephropathy. From this evidence, authors extend them conclusion considering that KLF2 may modulate fibrosis through the angiotensin type-1 receptor (AT1R) in hPGECs.

The In Vitro model of hypertension presented here and published by them previously (Experimental cell research 2020, 386 (1), 111706), is quite interesting. In addition, the hypothesis of the KLF2 role on renal fibrosis in hypertensive nephropathy is attractive, however, this study has several limitations where the quality of results is weak considering what is concluded in this work.

The Major commentaries are the following:

  • The 5/6 nephrectomy (Nx5/6) has been classically accepted as an experimental model of chronic kidney disease (CKD), and not as a model of hypertensive nephropathy. Even when authors use angiotensin II (AngII) to induce hypertension over the CKD model, I think that author should use only an experimental model with AngII infusion in high doses to induce hypertensive nephropathy. Nx5/6 induces general detrimental effects: inflammation, fibrosis, and proteinuria (Fig 5B), which does not occur in patients with hypertensive nephropathy (Table 2).  
  • Fig 1. After 48hrs of stimulation, authors may have an effect in KLF2 protein expression. The same for Fig.2. Why authors did not present the protein expression, by IF or IMHC for example, as they present them results in Fig6? I recommend completing these results.
  • Fig 3. Authors mention in tittle 3.3, a phenomenon on hPGECs that hey called as ‘pressure-induced fibrosis’. However, this is not correct since renal fibrosis is the histological manifestation where the hallmark of renal fibrosis, like all other organs, is pathological deposition of extracellular matrix. I recommend using ‘induction of fibrotic markers by pressure’ for cells. How is the alpha-SMA expression in this study with simvastatin and KLF2 siRNA? What is the efficacy of KLF2 siRNA? The authors only evaluated TGF-beta by mRNA. They should provide protein data and alpha-SMA regulation for different experimental conditions. Since they are using flow cytometry for these aims, I think that they may continue with the same technique for alfa-SMA and TGF-beta proteins.
  • Fig 4. The previous concern is the same for AT1R and AT2R (there is no protein analyzed).
  • Fig 4. Why they choose IL-18 and TNF-alpha as proinflammatory markers? What is the rationale behind it? Did they measure other proinflammatory mediators that promote fibrosis (e.g., IL-17, IL-23p19, IL-6, etc.). Maybe the explanation in discussion should be introduced before (introduction or during results description), in order to provide pertinency to these results. If author decide to study a specific role of AT1R, this must be probe by specific pharmacological antagonisms of receptors.
  • Fig 5. It is quite difficult to compare both groups in the representative figures for KLF2 considering the background differences between Sham and Nx5/6 It is necessary to present photos in similar quality. Here, it is not clear why authors did not study these effects in a model of AngII instead a Nx5/6 model.
  • The conclusion These KLF-2 mediated hypertensive damage results from increased IL-18 through AT1R’ is wrong from these results. There are no specific studies for AT1R or AT2R. In this sense, it would be interesting to determine if the pressure system by rotational force generates RAAS activation at cellular level (e.g., angiotensinogen/ACE overexpression). Moreover, considering that conclusion present edAT1R as critical in the effect of KLF2. By this way, it seems that the effect occurs by AngII more than HTA perse.

Reviewer 2 Report

In this manuscript, Bae et al. investigate the endothelial KLF2 roles in hypertensive  nephropathy with several models (human biopsy, in vitro primary culture of endothelial cells, in vivo 5/6 Unx).

The topic is interesting but there are some concerns about the study.

1) in Introduction, the authors must cite and explain the role of an other KFLs as KLF4 on endothelial integrity.

2) Animal model: The choice of the inbred strain C57BL/6 mouse, to induce hypertensive nephropathy model is to be discussed because this strain is relatively resistant to renal injury.

3) qRT-PCR: The geometric mean of multiple carefully selected housekeeping genes is necessary to interpret qPCR data.

4) No antibody references appear in the manuscript. Authors must specify it.

5) the statistical method is to be reviewed. the authors did not explain which test was used to analyze the data

6) there is a discrepancy between the values appearing in the contour plot and the values of the graphs especially for figure 2A. How do the authors explain this? 

7) To complete the analysis of apoptosis by another method than FACS the authors should perform immunoblot for cleaved caspase 3 and a tunel staining on culture cells for instance.

8)Cytokine analysis by qPCR is not sufficient. Authors should perform ELISA or FACS assays of cytokines particularly for IL18.

9) Fig 3B: the plot legend does not appear at the bottom. Please correct this mistake

10) In table 2 it would be interesting to specify treatments of patients (only the treatments that could have an impact on the results presented as anti-hypertensive therapies).

11) Fig.6: authors must show co-staining of an endothelial marker (as cD31) and KLF2.

Reviewer 3 Report

In their article entitled "Reno protective Effect of KLF2 on Glomerular Endothelial Dysfunction in Hypertensive Nephropathy", Bae et al provide a thorough characterization of the effect of KLF2 during hypertensive nephropathy. The structure of the experiments and the manuscript are sound and easy to follow. I think the article will be of interest to the readers of Cell.